# A Machine Learning Method for the Quality Detection of Base Liquor and Commercial Liquor Using Multidimensional Signals from an Electronic Nose

**DOI:** 10.3390/foods12071508

**Published:** 2023-04-03

**Authors:** Bingyang Li, Yu Gu

**Affiliations:** 1College of Information Science and Technology, Beijing University of Chemical Technology, Beijing 100029, China; 2School of Ocean Information Engineering, Jimei University, Xiamen 361021, China; 3Beijing Advanced Innovation Center for Soft Matter Science and Engineering, Beijing University of Chemical Technology, Beijing 100029, China

**Keywords:** Chinese liquor, electronic nose, residual network, light gradient boosting machine

## Abstract

Chinese liquor is a world-famous beverage with a long history. Base liquor, a product of liquor brewing, significantly affects the flavor and quality of commercial liquor. In this study, a machine learning method consisting of a deep residual network (ResNet)18 backbone with a light gradient boosting machine (LightGBM) classifier (ResNet-GBM) is proposed for the quality identification of base liquor and commercial liquor using multidimensional signals from an electronic nose (E-Nose). Ablation experiments are conducted to analyze the contribution of the framework’s components. Five evaluation metrics (accuracy, sensitivity, precision, F1 score, and kappa score) are used to verify the performance of the proposed method, and six other frameworks (support vector machine (SVM), random forest (RF), k-nearest neighbor (KNN), extreme gradient boosting (XGBoost), multidimensional scaling-support vector machine (MDS-SVM), and back-propagation neural network (BPNN)) on three datasets (base liquor, commercial liquor, and mixed base and commercial liquor datasets). The experimental results demonstrate that the proposed ResNet-GBM model achieves the best performance for identifying base liquor and commercial liquors with different qualities. The proposed framework has the highest F1 score for the identification of commercial liquor in the mixed dataset due to the contribution of similar microconstituents from the base liquor. The proposed method can be used for the quality control of Chinese liquor and promotes the practical application of E-nose devices.

## 1. Introduction

Chinese liquor is one of the most popular distillates worldwide and has a long history of over 6000 years [1]. According to the National Bureau of Statistics, approximately 7.2 billion liters of Chinese liquor was consumed in 2021, with sales of US$90 billion [2]. Chinese liquor is a traditional alcohol. It is generally produced from grains using traditional methods, including fermentation, distillation, storage, and blending [3]. The product obtained after distillation and storage without blending is called base liquor [4]. Almost all commercial Chinese liquors are blended using base liquors and specific blending techniques [5]. Although manufacturing processes differ, base liquors determine the quality of commercial liquors [6]. Due to the different base liquor qualities, there is a great variation of quality among different commercial liquors [7]. Therefore, evaluating base liquor quality is necessary for the quality control of commercial liquors.

The main components of Chinese liquors are alcohol and water, accounting for 98% of the total weight. Other components comprise less than 2% of the trace components but contribute to the complex aroma of the liquors, including esters, aldehydes, ketones, phenols, acids, nitrogen compounds, and sulfides [1]. The most common method for assessing the quality of base liquors is sensory evaluation and chemical/spectroscopic analysis [8]. However, the accuracy and objectivity of the results of the sensory evaluation method cannot be guaranteed because experts may be influenced by their health conditions, emotional states, or environmental factors [9]. Analysis methods, such as chromatography [10] and spectroscopy [11], are demanding and time-consuming. Base liquors that are inaccurately assessed are downgraded or destroyed, causing unnecessary waste. Therefore, it is necessary to develop an objective, convenient, rapid, and accurate method to detect the quality of base liquor.

An electronic nose (E-nose) is a device that simulates human olfactory perception using gas sensors. It consists of an array of gas sensors, a signal processing system, and a pattern recognition system [12]. Machine learning (ML) is a rapidly growing technology. It refers to algorithms that automatically learn information from data input [13]. Many researchers have distinguished Chinese commercial liquor using ML methods and E-nose data. Qi et al. [14] used an E-nose and support vector machine (SVM) classifier to distinguish six types of Chinese liquors with 90.8% accuracy. Jing et al. [15] employed an E-nose and a multilinear classifier to classify liquors with similar aromas and different prices, achieving an accuracy of 97.22%. Zhang et al. [16] proposed a channel attention convolutional neural network (CA-CNN) for the authenticity identification of Chinese liquor using an E-nose and obtained 98.53% prediction accuracy. Zhao et al. [17] presented a deep learning method with a stacked sparse autoencoder (SSAE) to classify seven brands of Chinese liquor utilizing E-nose data, achieving 96.67% prediction accuracy. Although the E-nose has shown good performance in distinguishing Chinese commercial liquors, few studies have focused on the classification of base liquors, which generally have similar microconstituents with very subtle differences, for different aging durations and aging environments. In addition, it is necessary to develop a method for the simultaneous identification of base and commercial liquors for practical applications.

In this study, we propose a novel ML framework for identifying base liquors and commercial liquors using multidimensional signals from an E-nose. The contributions of this paper can be summarized as follows.

A novel machine learning framework consisting of a deep residual network (ResNet)18 backbone and a light gradient boosting machine (LightGBM) classifier (ResNet-GBM) is proposed for the quality detection of base liquor and commercial liquor. The ResNet18 backbone is a powerful feature extractor and automatically extracts a sufficient number of comprehensive and significant features from the raw, multidimensional E-nose signals. The LightGBM is employed as the classifier to strengthen the identification ability of the liquor’s quality.Ablation and comprehensive comparative experiments are conducted on three datasets to analyze the contribution of the models’ components and the performance of the ResNet-GBM framework using five evaluation metrics (accuracy, sensitivity, precision, F1 score, and kappa score). A base liquor dataset and a commercial liquor dataset are used in the comparative experiments to assess the applicability of the proposed ResNet-GBM framework. In addition, a mixed dataset containing base liquor and commercial liquor data is used to evaluate the proposed framework’s robustness, generalization ability, and performance for the identification of Chinese liquors in a complex application scenario.The proposed method for the quality identification of base liquor and commercial liquor enables rapid detection and has high accuracy, providing a potential tool for quality control and promoting the practical application of E-nose devices.

## 2. Materials and Methods

### 2.1. Chinese Liquor Samples

All Chinese liquor samples (light flavor) were provided by the Shanxi Luxian Liquor Industry Co., Ltd. We tested nine types of Chinese liquor (six types of base liquors and three types of commercial liquors). The six types of base liquors with different aging durations were denoted as BL (year), where BL represents the base liquor, and (year) represents the aging duration. Thus, the six types of base liquors were BL (13), BL (11), BL (8), BL (6), BL (5), and BL (3). The details are listed in Table 1.

The three commercial liquors (CL) were blended using different proportions of the six base liquors. The details are listed in Table 2.

### 2.2. Instrument and Experiment

A PEN3 E-nose (Airsense Analytics GmbH, Schwerin, Germany) with a sensor array with ten metal oxide semiconductor (MOS) sensors (Table 3) was employed to collect the characteristic flavor information of the liquors.

All experiments were implemented in a clean and well-ventilated testing room of the authors’ laboratory at a temperature of 26 ± 2 °C and relative humidity of 50 ± 2%. The experiments lasted 25 days, and 9 different individual samples of each type were measured every day within the same procedure (total 25 days × 9 individual samples = 225 individual samples per type). These individual samples were from different production batches (the base liquor samples were from different liquor storages, and the commercial liquor samples were from different bottles) and were provided by the manufacturer directly. Each sample was measured once and updated if it was used, which ensured that no repeated measurements existed in the experiments. Therefore, the experiments contained 2025 independent measurements (9 types of liquor samples × 225 individual samples per type). Before the measurement, 3 mL of each sample was placed into a single hermetic vial (20 mL) and airproofed for 3 min to allow the liquor’s volatile compounds to disperse into the sampler.

The acquisition of the volatile compound profile was conducted in a well-ventilated location to minimize baseline fluctuations and interference from other volatile compounds. The zero gas (a baseline) was produced using two active charcoal filters (Filter 1 and Filter 2 in Figure 1) to ensure that the reference air and the air used for the samples had the same source.

The workflow of the E-nose includes the collection stage and flushing stage. Before collecting the data, an automatic zero-point trim was conducted for the E-nose by pumping clean air through filter 2 for 5 s. Then, the volatile compounds of the liquor sample were pumped into the sensor chamber with a flow rate of 600 mL/min to contact the sensor array for 100 s. During the collection stage, the gas molecules were adsorbed on the surface of the sensors, changing the sensors’ conductivity due to the redox reaction on the surface of the sensor’s active element. The sensors’ conductivity eventually stabilized at a constant value when the adsorption was saturated. The collection stage lasted 100 s, and sampling continued at one sample per second. In the flushing stage, clean air was pumped into the E-nose to flush the analytes. The collection and flushing stages were repeated to acquire the raw response data of the nine liquor samples. The workflow of the E-Nose followed the manufacturer’s instructions in the manual of the PEN3 E-nose [19].

### 2.3. Datasets

Three datasets (Dataset A, Dataset B, and Dataset C) were established using the multidimensional signals from the E-nose system. Dataset A was used to evaluate the performance of the proposed method for the classification of the six base liquors with different aging durations. Dataset B was utilized to test the effectiveness of the proposed method for the classification of commercial liquors with differing quality. Dataset C (mixed dataset) consisted of data from all liquor samples and was used to evaluate the ability of the proposed method to distinguish the base liquors and commercial liquors simultaneously.

Dataset A: This dataset comprised 1350 samples (25 days × 6 base liquors × 9 individual samples) of the six base liquors. The dataset for the SVM, random forest (RF), k-nearest neighbor (KNN), extreme gradient boosting (XGBoost), multidimensional scaling-support vector machine (MDS-SVM), and back-propagation neural network (BPNN) contained 27,000 samples (1350 samples × 20 measurements of the last 20 s) × 10 (number of sensors). The dataset for the ResNet-GBM framework contained 135,000 samples (1350 samples × 100 measurements during 100 s) × 10 sensors.

Dataset B: This dataset comprised 675 samples (25 days × 3 commercial liquors × 9 individual samples) of the three commercial liquors. The dataset for the SVM, RF, KNN, XGBoost, MDS-SVM, and BPNN contained 13,500 samples (675 samples × 20 measurements of the last 20 s) × 10 (number of sensors). The dataset for the ResNet-GBM framework contained 67,500 samples (675 samples × 100 measurements during 100 s) × 10 sensors.

Dataset C: This dataset comprised 2025 samples (25 days × 9 liquors × 9 individual samples) of the base liquors and commercial liquors. The dataset for the SVM, RF, KNN, XGBoost, MDS-SVM, and BPNN contained 40,500 samples (2025 samples × 20 measurements of the last 20 s) × 10 (number of sensors). The dataset for the ResNet-GBM framework contained 202,500 samples (2025 samples × 100 measurements during 100 s) × 10 sensors.

### 2.4. Principal Component Analysis

Principal component analysis (PCA) is a common multivariate statistical algorithm for dimensionality reduction (also known as feature extraction). It reduced the complexity of the data set while retaining most of the feature information [20]. The PCA accomplishes dimensionality reduction by transforming the original data into a new coordinate system according to the largest contribution of the variance from all variables; the coordinates are called the principal components [21]. Thus, a sample can be represented by a few principal components. A cumulative variance contribution of the first few principal components of 95% is considered reasonable [22].

### 2.5. Light Gradient Boosting Machine

The LightGBM is a learning framework based on the gradient boosting decision tree (GBDT) proposed by Microsoft Research in 2017 [23]. It has many improvements over the GBDT, such as gradient-based one-side sampling (GOSS) and exclusive feature bundling (EFB) to deal with data and features. It uses histogram-based algorithms to speed up the training process [24]. The LightGBM algorithm has the advantages of fast training speed, low memory overhead, no overfitting, and automatic feature processing. It also supports parallel processing and is suitable for large sample sizes and high-dimensional data [25].

### 2.6. ResNet

The ResNet is a commonly used convolutional neural network [26] consisting of a stack of residual blocks. It is not prone to gradient fading [27]. The residual block is shown in Figure 2, where x and y are the input and output of the block, respectively. $W_{1}$ and $W_{2}$ represent the weights of the first and second layers, respectively. The curvilinear arrows represent shortcut connections. $F\left(x\right)$ represents the first layer’s output after linear transformation and activation. After the linear transformation by the $W_{2}$ weight layer, $F\left(x\right)$ and the original input x are added to obtain $H\left(x\right)$, which is activated by a ReLU function to derive output y. Due to the residual blocks, ResNet can optimize the network layer, reducing redundancy [28].

### 2.7. Model Evaluation Metrics

Five evaluation metrics (*accuracy*, *sensitivity*, *precision*, *F*1 *score*, and *kappa score*) were used to assess model performance. Four parameters were used to calculate the metrics: True Positive (*TP*), False Positive (*FP*), True Negative (*TN*), and False Negative (*FN*).

*Accuracy* is defined as the proportion of correctly classified samples (*TP* samples and *TN* samples) to the total number of samples; it is calculated as follows:(1)$$Accuracy=\frac{TP+TN}{TP+TN+FP+FN}$$

*Sensitivity* represents the proportion of the *TP* samples to the total number of positive samples (*TP* samples and *FN* samples); it is defined as follows:(2)$$Sensitivity=\frac{TP}{TP+FN}$$

*Precision* (*P*) represents the proportion of the *TP* samples to the total number of positive predictions (*TP* samples and *FP* samples); the formula is as follows:(3)$$Precision=\frac{TP}{TP+FP}$$

The *F1 score* is defined as the harmonic mean of the *precision* and *sensitivity*; it is calculated as follows:(4)$$F1=\frac{2*Precision*Sensitivity}{Precision+Sensitivity}$$

The *kappa score* is an evaluation metric for multi-class classification models that measures the consistency between categories and the classification accuracy: it is calculated as follows:(5)$$p_{0}=\frac{TP+TN}{TP+TN+FP+FN}$$
(6)$$p_{e}=\frac{\left(TP+FN\right)\left(TP+FP\right)\left(TN+FN\right)\left(TN+FP\right)}{{\left(TP+TN+FP+FN\right)}^{2}}$$
(7)$$kappa=\frac{p_{0}-p_{e}}{1-p_{e}}$$
where $p_{0}$ represents the overall classification accuracy, and $p_{e}$ is the expected agreement.

## 3. Proposed Method

An ML framework called ResNet-GBM, consisting of a ResNet18 backbone and a LightGBM classifier, was proposed to process the multidimensional signals from the E-Nose. Figure 3 shows the flowchart of ResNet-GBM.

The raw data obtained from the 10 channels was a 100 (measurement time of 100 s) × 10 (number of MOS sensors) matrix. Due to the multi-channel input of the ResNet18 model, a 10-channel input was constructed by converting the raw E-nose data from the sensor array. The 100 raw response points of each sensor were converted into a 10 × 10 matrix, and the 10 matrices of the sensor array were converted into a 10 × 10 × 10 matrix used for the 10-channel input of the ResNet18 model.

As shown in Figure 4c, the ResNet18 backbone consists of six stages, including Conv 1, Layer 1, Layer 2, Layer 3, Layer 4, and Pool. Conv 1 consists of a convolutional layer, a batch normalization layer, a ReLU layer, and a pooling layer. The kernel size, stride size, and padding size of the convolutional layer are 7 × 7, 2, and 3, respectively. The kernel size, stride size, and padding size of the pooling layer are 3 × 3, 2, and 1, respectively. Layer 1 consists of two Basic Blocks 1. Layer 2, Layer 3, and Layer 4 consist of Basic Block 1 and Basic Block 2. The two Basic Blocks are presented in Figure 4a,b.

There are two convolutional layers in Basic Block 1, and the kernel size, stride size, and padding size of the two convolutional layers have the same value: 3 × 3, 1, and 1, respectively. Basic Block 2 has three convolutional layers, two of which are down-sampling layers. The kernel size, stride size, and padding size of the down-sampling convolutional layer 1 are 3 × 3, 2, and 1, respectively. The kernel size, stride size, and padding size of the down-sampling convolutional layer 2 are 1 × 1, 2, and 1, respectively. The kernel size, stride size, and padding size of the third convolutional layer are 3 × 3, 1, and 1, respectively. The details of the ResNet18 backbone are listed in Table 4.

The 512 features extracted by the ResNet18 backbone were input into the LightGBM model. In the LightGBM, a grid search method (GSM) was employed to derive the main parameters (num_leaves and learning_rate) and obtain the best performance of the classifier. After training the proposed model using the training set, the test set was used to evaluate the effectiveness of the trained model for the quality detection of base liquors and commercial liquors.

## 4. Results and Discussion

### 4.1. Principal Component Analysis

Since the measurement phase lasted 100 s, and the response value of each sensor was stable after 80 s, the last 20 response points were chosen as the input features for PCA. As shown in Figure 5, the x-axis, y-axis, and z-axis represent principal component 1 (PC1), principal component 2 (PC2), and principal component 3 (PC3), respectively. The percentages of the variance of PC1, PC2, and PC3 were 69.4%, 15.3%, and 11.5%, respectively. The cumulative variance of PC1, PC2, and PC3 was 96.2%, indicating that sufficient sample information was contained in the three principal components. However, the subplot indicated that the clusters were in close proximity, i.e., there was overlap. During dimensionality reduction, some principal components with a small contribution rate were overlooked, but these components may have contained critical information. Therefore, PCA is not an effective method for separating the classes in this study.

### 4.2. Experiments

Four sets of experiments were conducted: experiment I, experiment II, experiment III, and experiment IV. Experiment I was an ablation study to verify the contributions of the proposed ResNet-GBM’s components. Experiments II to IV were performed to compare the proposed ResNet-GBM framework with six other methods (including four common machine learning methods (SVM, RF, KNN, and XGBoost) and two methods proposed by other authors (MDS-SVM and BPNN)) on Dataset A, Dataset B, and Dataset C, respectively. MDS-SVM is a pattern recognition method based on multidimensional scaling and SVM. It was developed by Li et al. [29] to classify ten brands of Chinese liquors. BPNN is a multi-layered feedforward neural network that was used by Liu et al. [30] to distinguish different wines based on their properties. The models were implemented using NVIDIA GeForce MX250 graphics cards and the open-source PyTorch framework.

#### 4.2.1. Experiment I: Ablation Study of ResNet-GBM

The comparative results of the ablation experiments on Datasets A, B, and C are displayed in Table 5. The proposed ResNet-GBM had the best performance for the classification of the Chinese liquor on the three datasets with accuracies of 0.9704, 0.9814, and 0.9803, respectively. The LightGBM provided an unsatisfactory performance and could not identify the liquors accurately. It is possible that it cannot mine a sufficient number of features. The ResNet18 exhibited better performance than the LightGBM, indicating that the automatic feature extraction capability significantly improved the classification performance. The proposed ResNet-GBM model combines the merits of ResNet18 and the LightGBM classifier, reducing the possibility of overfitting during training and improving the model’s classification performance.

#### 4.2.2. Experiment II: Performance of the Proposed Framework on Dataset A

Dataset A was used in Experiment II. It contains 6 types of base liquor with different aging durations. Dataset A was divided into training sets (data from the first twenty days: 20 days × 6 liquor samples × 9 individual samples) and test sets (data from the last five days: 5 days × 6 liquor samples × 9 individual samples).

The radial basis function (RBF) was selected as the kernel function of the SVM model. The SVM model includes two important parameters: the penalty coefficient (C) and kernel function coefficient (gamma). A GSM with C ∈ [10, 50, 100, 200, 500] and gamma ∈ [0.1, 1.0, 5.0, 10.0, 20.0] was employed to determine the optimal parameters. The C was 100, and the gamma was 1. The number of decision trees and the number of randomly selected features obtained from each decision tree ($N_{F}$) in the RF model were selected using the equation $N_{F}=\sqrt{M}$, where M represents the number of features. In our experiments, 10 decision trees ($N_{F}$ = 10) were used. In the KNN model, the *n*_neighbors is a critical parameter; it was set to 3. The parameters of the XGBoost model were similar to those of the LightGBM model. The number of leaves (num_leaves) and the learning rate (learning_rate) are important parameters of the LightGBM model. The GSM was employed to search the parameters; the range of values for num_leaves was [8, 16, 32, 64, 128], and the learning_rate was [0.001, 0.01, 0.1, 0.5, 1]. The num_leaves was 16, and the learning_rate was 0.1.

The classification results derived from the seven models are displayed in Table 6. Five evaluation metrics were used to evaluate the classification models. As shown in Table 6, the proposed ResNet-GBM obtained the best performance for all evaluation metrics, with an accuracy of 0.9704, a sensitivity of 0.9704, a precision of 0.9716, an F1 score of 0.9710, and a kappa score of 0.9644. SVM, RF, KNN, XGBoost, MDS-SVM, and BPNN achieved accuracies of 0.3175, 0.4018, 0.4053, 0.4246, 0.7852, and 0.8963, respectively. The performances of these six models were unsatisfactory because they failed to extract a sufficient number of deep features. The experimental results demonstrated the effectiveness and superior performance of the proposed ResNet-GBM framework to identify different base liquors.

#### 4.2.3. Experiment III: Performance of the Proposed Framework on Dataset B

Experiment III used commercial liquor samples to assess the generalization performance of the models. Dataset B was divided into training sets (data from the first twenty days: 20 days × 3 liquor samples × 9 individual samples) and test sets (data from the last five days: 5 days × 3 liquor samples × 9 individual samples).

The parameters of the models were the same as those in Experiment II. The results are listed in Table 7. The ResNet-GBM model exhibited the best results for the commercial liquors, with an accuracy of 0.9814, a sensitivity of 0.9814, a precision of 0.9825, an F1 score of 0.9815, and a kappa score of 0.9722. SVM, RF, KNN, XGBoost, MDS-SVM, and BPNN achieved accuracies of 0.3649, 0.6351, 0.5088, 0.6105, 0.8235, and 0.9118, respectively. The results showed that the proposed model could accurately detect different grades of commercial liquor. The classification performance of the model was better for the commercial liquor than for the base liquor.

#### 4.2.4. Experiment IV: Performance of the Proposed Framework on Dataset C

Experiment IV evaluated the generalization performance of the ResNet-GBM model on Dataset C (a mixture of base liquors and commercial liquors). Dataset C was also divided into training sets (data from the first twenty days: 20 days × 9 liquor samples × 9 individual samples) and test sets (data from the last five days: 5 days × 9 liquor samples × 9 individual samples).

The classification results of the seven models are listed in Table 8. The ResNet-GBM framework obtained the best results with an accuracy of 0.9803, a sensitivity of 0.9803, a precision of 0.9819, an F1 score of 0.9801, and a kappa score of 0.9778 for the simultaneous classification of the base liquors and commercial liquors. SVM, RF, KNN, XGBoost, MDS-SVM, and BPNN achieved accuracies of 0.3389, 0.4468, 0.4304, 0.4901, 0.8148, and 0.9074, respectively. The comparison results indicate that the ResNet-GBM framework achieved better performances for extracting significant features from the multidimensional sensor signals and provided superior performance for the classification of base liquors, commercial liquors, and a mixture of both.

We further analyzed the results of each sample and selected the comprehensive evaluation metric F1 score to assess the performance of ResNet-GBM. The results are listed in Table 9. The proposed model achieved similar performances in Experiment II and Experiment IV for the classification of the base liquors (BL13, BL11, BL8, BL6, BL5, and BL3). The F1 scores of CL1 and CL2 in Experiment IV are 1.0000, higher than that in Experiment III. The comparison results showed that the classification performance of the proposed model for commercial liquor was higher when base liquors and commercial liquors were analyzed simultaneously. This result indicated that the model could mine deeper features based on the base liquor samples, which contributed to the high classification accuracy of the commercial liquors in Experiment IV.

## 5. Conclusions

We proposed a ResNet-GBM framework to identify base liquors and commercial liquors with different qualities using a MOS-based E-nose. The main conclusions are as follows:

PCA was used to distinguish nine liquor samples using the E-nose data. High coincidence points in the PCA result indicated that the odor information of different liquors was highly similar. The unsatisfactory PCA results indicated that this method could not distinguish liquors with different qualities, and meaningful feature information was lost during dimensionality reduction.

A ResNet-GBM framework consisting of the ResNet18 backbone and the LightGBM classifier was proposed for the quality detection of base liquors and commercial liquors. Ablation experiments were conducted to determine the contributions of the ResNet-GBM’s components for identification. The results indicated the effectiveness of the proposed framework. The significant features contained in the multidimensional signals were extracted by the ResNet18 backbone. The LightGBM classifier strengthened the identification ability of the ResNet model, and the proposed model achieved classification accuracies of 0.9704, 0.9814, and 0.9803 for Datasets A, B, and C, respectively.

The superiority of the proposed framework was demonstrated by comparing it with six other methods (SVM, RF, KNN, XGBoost, MDS-SVM, and BPNN) using the three datasets. The comparative experiments proved that the proposed framework had higher classification performance and better generalization ability than the other models using the multidimensional E-nose signals as input.

The F1 scores of the ResNet-GBM model for all samples were compared using the three datasets (base liquor dataset, commercial liquor dataset, and mixed dataset). The proposed ResNet-GBM model achieved better performance for the classification of commercial liquor using the mixed dataset (1.0000 for CL1, CL2, and CL3) than the commercial liquor dataset (0.9730 for CL1, 0.9714 for CL2, and 1.0000 for CL3). The results indicated that the excellent performance for distinguishing base liquors resulted in a higher classification accuracy of commercial liquors when base liquors and commercial liquors were analyzed simultaneously.

The results were encouraging and demonstrated that a deep learning framework could be used to identify base liquors and commercial liquors with different qualities using E-nose data. This approach provides a potential tool for the quality control of liquor and promotes the practical application of E-nose devices. This deep learning framework is expected to have broad application value for food quality control.

## Figures and Tables

**Figure 1 foods-12-01508-f001:**
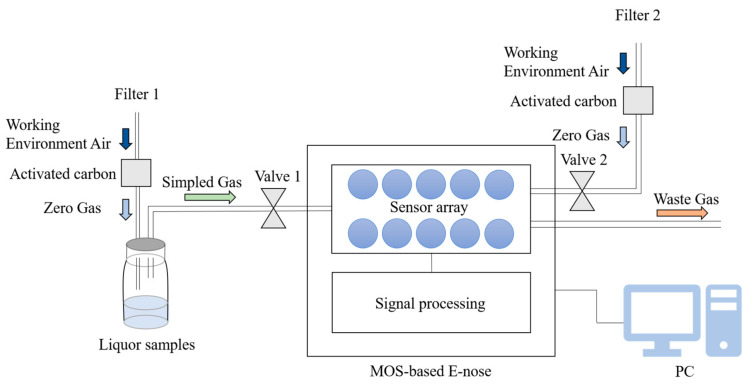
Schematic diagram of the PEN-3 workflow.

**Figure 2 foods-12-01508-f002:**
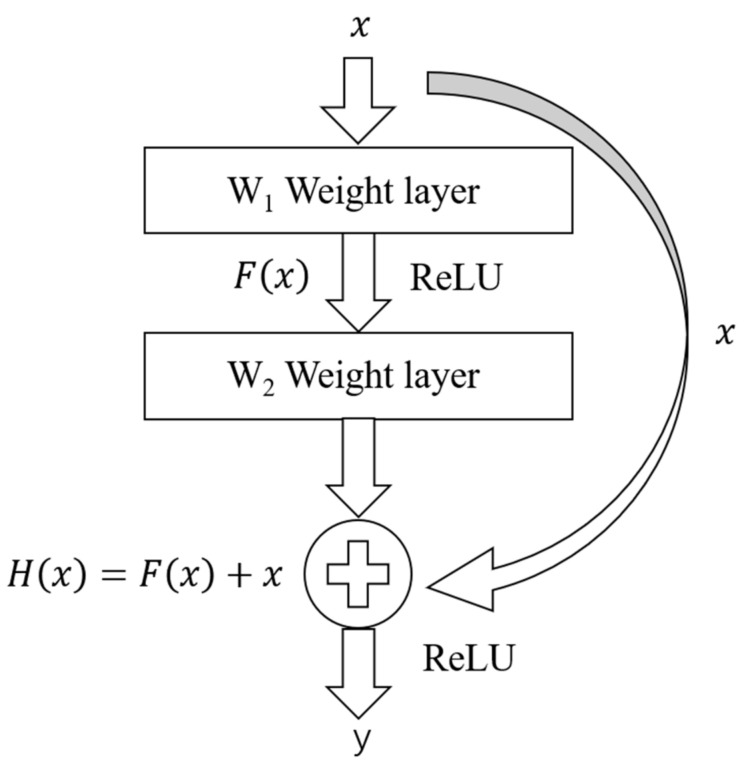
The residual block.

**Figure 3 foods-12-01508-f003:**
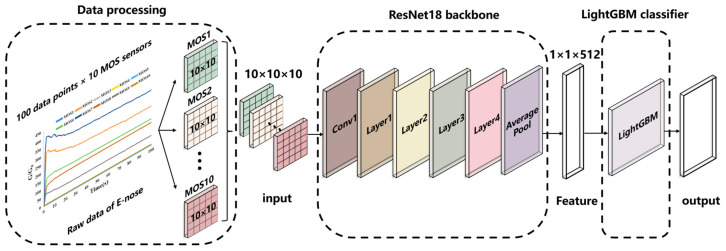
The classification framework for the quality detection of Chinese liquor.

**Figure 4 foods-12-01508-f004:**
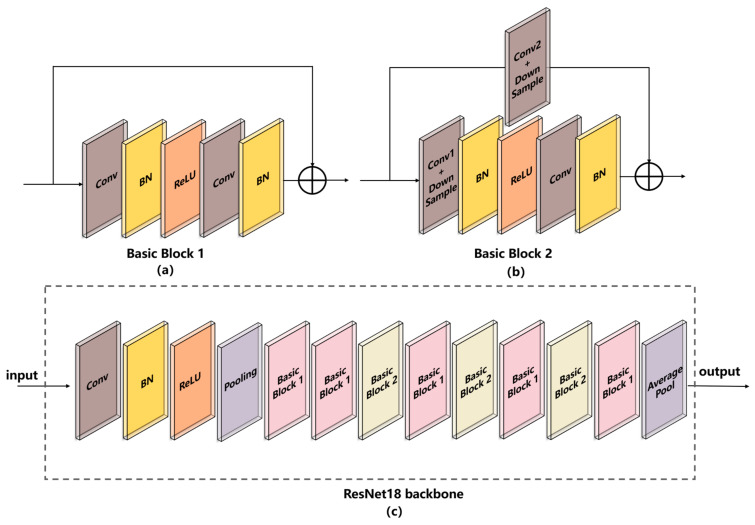
Schematic diagram of the ResNet18 backbone. (**a**) Basic Block 1; (**b**) Basic Block 2; (**c**) ResNet18 backbone.

**Figure 5 foods-12-01508-f005:**
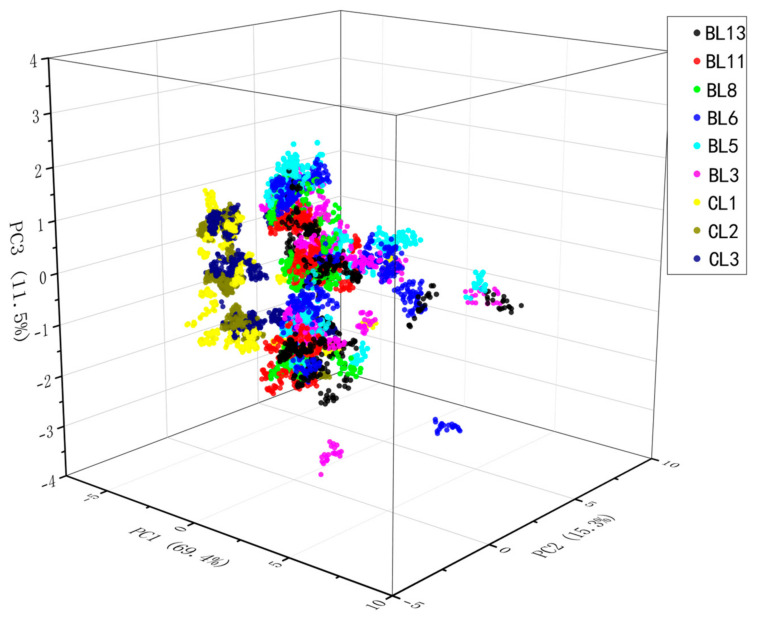
PCA score plot of the liquor samples with the first three principal components.

**Table 1 foods-12-01508-t001:** Details of the base liquor.

Label	Aging Duration	Alcohol Content
BL13	13	65
BL11	11	65
BL8	8	65
BL6	6	65
BL5	5	65
BL3	3	65

**Table 2 foods-12-01508-t002:** Details of the commercial liquors.

Label	Production Year	Alcohol Content	Components	Blending Proportion (%)
CL1	2022	42	BL13	0.01
			BL11	8.99
			BL5	31
			BL3	60
CL2	2022	42	BL13	0.008
			BL8	5.992
			BL5	24
			BL3	70
CL3	2022	42	BL13	0.006
			BL6	3.994
			BL5	16
			BL3	80

**Table 3 foods-12-01508-t003:** Standard sensor array of the PEN3 E-nose [18].

Number	Sensor	Main Performance
MOS1	W1C	Aromatic constituent
MOS2	W5S	Nitride oxides
MOS3	W3C	Ammonia and aromatic constituent
MOS4	W6S	Hydrogen
MOS5	W5C	Alkanes and aromatic constituent
MOS6	W1S	Methane
MOS7	W1W	Sulfide
MOS8	W2S	Alcohol
MOS9	W2W	Aroma constituent and organic sulfur compounds
MOS10	W3S	Alkanes

**Table 4 foods-12-01508-t004:** Details of the ResNet18 backbone.

Stage	Output	Structure Details
Conv1	112 × 112 × 64	7 × 7, 64, s = 2, *p* = 3
	56 × 56 × 64	3 × 3, max-pooling, s = 2, *p* = 1
Layer1	56 × 56 × 64	$\left[\begin{array}{c} 3\times 3.64\\ 3\times 3.64 \end{array}\right]$×2
Layer2	28 × 28 × 128	$\left[\begin{array}{c} 3\times 3.128\\ 3\times 3.128 \end{array}\right]$×2
Layer3	14 × 14 × 256	$\left[\begin{array}{c} 3\times 3.256\\ 3\times 3.256 \end{array}\right]$×2
Layer4	7 × 7 × 512	$\left[\begin{array}{c} 3\times 3.512\\ 3\times 3.512 \end{array}\right]$×2
Pool	1 × 1 × 512	Global average pooling

**Table 5 foods-12-01508-t005:** Results of ablation experiment.

Test Dataset		Dataset A	Dataset B	Dataset C
LightGBM	Accuracy	0.4053	0.5088	0.4304
	Sensitivity	0.4053	0.5088	0.4304
	Precision	0.3732	0.5287	0.4141
	F1 score	0.3768	0.5150	0.4138
	Kappa score	0.2863	0.4789	0.3592
ResNet18	Accuracy	0.9037	0.9185	0.9074
	Sensitivity	0.9037	0.9185	0.9074
	Precision	0.9238	0.9274	0.9181
	F1 score	0.9064	0.9183	0.9084
	Kappa score	0.8844	0.9022	0.8889
ResNet-GBM	Accuracy	**0.9704**	**0.9814**	**0.9803**
	Sensitivity	**0.9704**	**0.9814**	**0.9803**
	Precision	**0.9716**	**0.9825**	**0.9819**
	F1 score	**0.9710**	**0.9815**	**0.9801**
	Kappa score	**0.9644**	**0.9722**	**0.9778**

**Table 6 foods-12-01508-t006:** Classification results of seven models in Experiment II.

Model	Accuracy	Sensitivity	Precision	F1 Score	Kappa Score
SVM	0.3175	0.3175	0.3531	0.3042	0.1811
RF	0.4018	0.4018	0.3696	0.3590	0.2821
KNN	0.4053	0.4053	0.3732	0.3768	0.2863
XGBoost	0.4246	0.4246	0.4437	0.4096	0.3095
MDS-SVM [28]	0.7852	0.7826	0.8206	0.7740	0.7419
BPNN [29]	0.8963	0.8962	0.8995	0.8948	0.8755
ResNet-GBM	**0.9704**	**0.9704**	**0.9716**	**0.9710**	**0.9644**

**Table 7 foods-12-01508-t007:** Classification results of seven models in Experiment III.

Model	Accuracy	Sensitivity	Precision	F1 Score	Kappa Score
SVM	0.3649	0.3649	0.3677	0.3501	0.2379
RF	0.6351	0.6351	0.6453	0.6322	0.5237
KNN	0.5088	0.5088	0.5099	0.5086	0.4912
XGBoost	0.6105	0.6105	0.6767	0.6165	0.5064
MDS-SVM [28]	0.8235	0.8202	0.8737	0.8206	0.7345
BPNN [29]	0.9118	0.9130	0.9183	0.9115	0.8677
ResNet-GBM	**0.9814**	**0.9814**	**0.9825**	**0.9815**	**0.9722**

**Table 8 foods-12-01508-t008:** Classification results of seven models in Experiment IV.

Model	Accuracy	Sensitivity	Precision	F1 Score	Kappa Score
SVM	0.3389	0.3389	0.3299	0.3268	0.2563
RF	0.4468	0.4468	0.4764	0.4290	0.3776
KNN	0.4304	0.4304	0.4141	0.4139	0.3592
XGBoost	0.4901	0.4901	0.5337	0.4849	0.4263
MDS-SVM [28]	0.8148	0.8148	0.8420	0.8161	0.7917
BPNN [29]	0.9074	0.9074	0.9170	0.9074	0.8958
ResNet-GBM	**0.9803**	**0.9803**	**0.9819**	**0.9801**	**0.9778**

**Table 9 foods-12-01508-t009:** Results of F1 scores for each sample obtained from ResNet-GBM in Experiments II, III, and IV.

Label	Experiment II	Experiment III	Experiment IV
BL13	0.9778	n.e.	0.9778
BL11	0.9524	n.e.	0.9545
BL8	0.9565	n.e.	0.9565
BL6	0.9787	n.e.	0.9778
BL5	0.9565	n.e.	0.9787
BL3	0.9645	n.e.	0.9767
CL1	n.e.	0.9730	**1.0000**
CL2	n.e.	0.9714	**1.0000**
CL3	n.e.	1.0000	**1.0000**

n.e.: not existent

## Data Availability

The data presented in this study are available at Figshare (https://doi.org/10.6084/m9.figshare.22351597.v1 (accessed on 29 March 2023)).

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
