# Peer review of "A Machine Learning Method for the Quality Detection of Base Liquor and Commercial Liquor Using Multidimensional Signals from an Electronic Nose"

_foods, 2023, doi:10.3390/foods12071508_

Round 1

Reviewer 1 Report (Previous Reviewer 3)

I recommend the paper publishing.

Author Response

Thank you very much for your comment

Reviewer 2 Report (Previous Reviewer 2)

Dear Authors,

Thank you for accepting the suggested edits and corrections. I think it helped improve your manuscript.

Author Response

Thank you very much for your comment

Reviewer 3 Report (Previous Reviewer 1)

I have reviewed previous version of this manuscript and proposed changes were not implemented in the re-submitted version, which is in fact a copy of the previous one. The main issue of the work is a very small number of independent samples, which makes impossible application of the classification method proposed by the authors. I can only repeat my comments for the last time:

The main problem of this work is limited number of analysed samples. There is a basic misunderstanding on what constitutes an independent sample. Several measurements in the same samples, made on the same or on different days, are not independent samples. These are replicated measurements. Chemical composition of a sample remains exactly the same independently of how many times it was analyzed. The difference observed between these replicated measurements are due to the sensors' drift. Thus, in this work classification models were calculated using 1 sample per class, which is clearly inadequate. If another liquor sample belonging e.g. to the class BL13 is presented to the classification model, it would not be classified correctly as there are slight differences in composition between liquor samples of the same type.

If analytical instrument used for measurements was ideal, its response in this sample would be exactly the same in all repeated measurements. But in reality, there are always differences in the instrumental response due to the drift, and in the case of e-nose, as PCA score plot demonstrates, these differences are quite significant, which could be expected as e-noses are known to be prone to drift.

The large data set acquired in this work could be used for study of the drift reduction / recalibration approaches, which requires reformulation of the machine learning method. Alternatively, the authors can apply proposed classification approach to the other data set.

Author Response

The main problem with this work is limited number of analysed samples. There is a basic misunderstanding on what constitutes an independent sample. Several measurements in the same sample, made on the same or on different days, are not independent samples. These are replicated measurements. Chemical composition of a sample remains exactly the same independently of how many times it was analyzed.

If analytical instrument used for measurements was ideal, its response in the same sample would be exactly the same in all repeated measurements. But in reality, there are always differences in the instrumental response due to the drift, and in the case of e-nose, as PCA score plot demonstrates, these differences are quite significant, which could be expected as e-noses are prone to drift.

In this work classification model was calculated using 1 sample per class, which is clearly inadequate. If another liquor sample belonging to e.g. the class BL13 is presented to such classification model, it would be misclassified as there are slight differences in composition between liquor samples of the same type.

The large data set acquired in this work could be used for study of the drift reduction / recalibration approaches, which would require reformulation of the machine learning method. Alternatively, the authors can apply proposed classification approach to the other data set.

Author response: Dear reviewer, thanks a lot for your attentions and suggestions.

We totally agree with your opinion about the constitution of independent sample. Actually, in our study, the design of our experiments was same with your minds that all the measurements were absolutely on the independent samples. However, we did not clearly describe the details of the experiment samples in our manuscript, which caused your misunderstanding. We are also sorry that we did not answer the question of the samples in our study so clearly in response (Round 1). Now, our explanations are as follows:

  1. As you said, several measurements in the same sample, made on the same or on different days, are not independent samples. As we introduced in our manuscript, in our study, all the liquor samples were provided by the Shanxi Luxian Liquor Industry Co., Ltd. The experiments lasted for 25 days, and six types of base liquors and three types of commercial liquors were prepared for the experiments (here, we point out that nine types of liquor samples). As regards each type of liquors, nine different individual samples (the base liquors were not from the same liquor storage, and the commercial liquors were not from the same bottle) with same volume (10ml) were provided everyday by the manufacturer and measured using E-nose (total 25 days × 9 individual samples = 225 individual samples per type). Therefore, the total number of samples in our study was 2025 (9 types of liquor samples × 225 individual samples per type). Each sample was measured once and updated if it was used, which ensures that no repeated measurements exist in the experiments. Therefore, the measurements on the same or on different days were not in the same samples (replicated measurements), but in the independent samples (independent measurements). According to your suggestions, we have revised the experiment description in Section 2.2. Instrument and Experiment (Line120-129, Page 4, Clean revised version).

“…The experiments lasted for 25 days, and nine different individual samples of each type were measured every day within the same procedure (total 25 days × 9 individual samples = 225 individual samples per type). These individual samples were from different production batches (the base liquor samples were from different liquor storages, and the commercial liquor samples were from different bottles) and provided by the manufacturer directly. Each sample was measured once and updated if it was used, which ensured that no repeated measurements existed in the experiments. Therefore, the experiments contained 2025 independent measurements (9 types of liquor samples × 225 individual samples per type) ….”

  1. In our study, our aim is to propose a machine learning method for the quality detection of base liquor and commercial liquor using multidimensional signals from an MOS-based E-nose. As far as we known, in the researches of food quality detection by means of an E-nose system, almost all the researchers conducted the experiments using samples from different batches and some researchers even conducted repeat measurements on the samples from the same batch. Zhang et al. [1] proposed a channel attention convolutional neural network for the detection of 10 kinds of Chinese liquor, with each liquor being presented in six different production batches. As regards each type of liquors in different batches, 20 samples with 50ul were obtained by reusing the dropper (total 6 batches × 20 measurement times = 120 samples per kind). Li et al. [2] used an E-nose to recognize odors and diagnose quality of bacon. Five different batches of manufactured bacon were sampled and for each production batch 18 parallel samples were selected randomly, a total of 90 samples of smoked bacon were tested in experiments (total 5 batches × 18 measurement times = 90 samples per kind). Xu et al. [3] proposed a rapid detection method of jointly using E-nose and computer vision system to detect tea aroma and tea appearance for tea quality identification. And 32 replicates for each tea grade were detected by the E-nose (32 samples per kind). Therefore, we think that the samples (total 25 days × 9 individual samples = 225 individual samples per kind) in our study is enough for the E-nose experiments. Therefore, we think that the experiment in our study is workable and conformed to standards.
  2. As you said, there are always differences in the instrumental response due to the drift, which could be expected as e-noses are prone to drift. As we all known, the drift is a rather complex and inevitable effect in real scenarios, which is generated by different sources (such as sensor aging, sensor poisoning, thermal and memory effects of sensors, changes in environment and odor delivery system noise) [4]. So, strictly speaking, it is almost impossible to acquire a data set without any drift in real scenarios. In addition, the aim of our study is proposing a rapid and accurate method for the quality detection of base liquors and commercial liquors by means of a E-nose system in real scenarios. So, we conducted the experiments by independent measurements on individual samples in 25 days to satisfy the conditions of E-nose applications in real scenarios as much as possible. As a result, the distribution of experiment data in our study was closer to fact. PCA was used to distinguish the nine liquor samples as a regular tool in our study, and the results showed that PCA was not capable in this case. Therefore, we proposed a novel framework, namely ResNet-GBM, to mine the deep aroma features of the liquors for the quality detection of the nine liquor samples. The experimental results indicated that our proposed model presented good performances and strong robustness for the classification of the nine Chinese liquors, which shows a potential to avoiding the negative effect of the time drift in E-nose.
  3. In our study, the experiments were performed for 25 days to obtain more data, and the individual samples were measured to obtain more representative data. The data in the first 20 days was used for training (total 20 days × 9 individual samples = 180 individual samples per kind) and that in the last 5 days was used for test (total 5 days × 9 individual samples = 45 individual samples per kind). So, the training dataset and test dataset are independent, which can be used as a proper validation for the assessment of classification models.

References:

[1] Zhang, S., Cheng, Y., Luo, D., He, J., Wong, A. K., & Hung, K. (2021). Channel attention convolutional neural network for Chinese baijiu detection with E-nose. IEEE Sensors Journal, 21(14), 16170-16182.

[2] Li, X., Zhu, J., Li, C., Ye, H., Wang, Z., Wu, X., & Xu, B. (2018). Evolution of volatile compounds and spoilage bacteria in smoked bacon during refrigeration using an E-Nose and GC-MS combined with partial least squares regression. Molecules, 23(12), 3286.

[3] Xu, M., Wang, J., & Gu, S. (2019). Rapid identification of tea quality by E-nose and computer vision combining with a synergetic data fusion strategy. Journal of Food Engineering, 241, 10-17.

[4] Ziyatdinov, A., Marco, S., Chaudry, A., Persaud, K., Caminal, P., & Perera, A. (2010). Drift compensation of gas sensor array data by common principal component analysis. Sensors and Actuators B: Chemical, 146(2), 460-465.

This manuscript is a resubmission of an earlier submission. The following is a list of the peer review reports and author responses from that submission.

Round 1

Reviewer 1 Report

The manuscript by Bingyang Li et al., titled “A Machine Learning Method for the Quality Detection of Base Liquor and Commercial Liquor Using Multidimensional Signals from an Electronic Nose” reports on the application of a deep-learning neural network for classification of Chinese liquor using e-nose measurements. Application of machine learning techniques to the multisensory data is an interesting area of research, however, approach presented in this work has serious drawbacks. First of all, despite of a large number of measurement points collected, number of samples is very small. For instance, models for discrimination of base and commercial liquors are made with 6 and 3 samples, respectively. Given that classification models are calculated using large neural networks, these models are bound to be overfitted, especially because a proper validation of the classification models is impossible to implement. It is mentioned (lines 316-317) that data set was split in the training and testing data set, but in this case both data sets included measurements in the samples made on different days. PCA score plot presented in the Fig. 5 does not provide assurance that e-nose is capable to distinguish liquor samples as they overlap. What PCA score plot does show is that there is significant drift in the data. Considering this, the data sets collected in this work could be more suitable for the study of drift reduction than to develop classification models.

Comments.

1) Please provide a table with analyzed base liquor listed, their age, and alcohol content of all base and commercial liquors analyzed.

2) Ethanol is a major compound of liquors, and it can influence sensor response. Have sensors’ sensitivity to ethanol been studied?

3) Chinese liquor composition has been extensively studied. Please provide some overview of the main compounds that are believed to contribute to the liquor quality.

4) What was the rational for sensor selection for an e-nose? For examples, sensors sensitive primarily to hydrogen (W6S) and alkanes (W3S) were included in the sensor array, though obviously these compounds are not expected to be present in the liquor headspace.

5) Table 3 is unnecessary, explanation given in the text is sufficient.

6) Recall and precision are terms used for sensitivity and specificity in machine learning. As the Foods is a food science and technology journal, I would suggest to use terminology with which its readers are more familiar, i.e. sensitivity and specificity.

7) Classification results presented in the tables 5-9 refer to train or test data set?

Author Response

Original Manuscript ID: foods-2204718     

Original Article Title: “A Machine Learning Method for the Quality Detection of Base Liquor and Commercial Liquor Using Multidimensional Signals from an Electronic Nose

Reviewer #1 (Comments and Suggestions for Authors):
The manuscript by Bingyang Li et al., titled “A Machine Learning Method for the Quality Detection of Base Liquor and Commercial Liquor Using Multidimensional Signals from an Electronic Nose” reports on the application of a deep-learning neural network for classification of Chinese liquor using e-nose measurements. Application of machine learning techniques to the multisensory data is an interesting area of research, however, approach presented in this work has serious drawbacks. First of all, despite of a large number of measurement points collected, number of samples is very small. For instance, models for discrimination of base and commercial liquors are made with 6 and 3 samples, respectively. Given that classification models are calculated using large neural networks, these models are bound to be overfitted, especially because a proper validation of the classification models is impossible to implement. It is mentioned (lines 316-317) that data set was split in the training and testing data set, but in this case both data sets included measurements in the samples made on different days. PCA score plot presented in the Fig. 5 does not provide assurance that e-nose is capable to distinguish liquor samples as they overlap. What PCA score plot does show is that there is significant drift in the data. Considering this, the data sets collected in this work could be more suitable for the study of drift reduction than to develop classification models.

Comments.

1) Please provide a table with analyzed base liquor listed, their age, and alcohol content of all base and commercial liquors analyzed.

2) Ethanol is a major compound of liquors, and it can influence sensor response. Have sensors’ sensitivity to ethanol been studied?

3) Chinese liquor composition has been extensively studied. Please provide some overview of the main compounds that are believed to contribute to the liquor quality.

4) What was the rational for sensor selection for an e-nose? For examples, sensors sensitive primarily to hydrogen (W6S) and alkanes (W3S) were included in the sensor array, though obviously these compounds are not expected to be present in the liquor headspace.

5) Table 3 is unnecessary, explanation given in the text is sufficient.

6) Recall and precision are terms used for sensitivity and specificity in machine learning. As the Foods is a food science and technology journal, I would suggest to use terminology with which its readers are more familiar, i.e. sensitivity and specificity.

7) Classification results presented in the tables 5-9 refer to train or test data set?

Dear reviewer,

Thanks for your supporting. Our answers for your questions are as follows:

Reviewer #1, Concern # 1:

- First of all, despite of a large number of measurement points collected, number of samples is very small. For instance, models for discrimination of base and commercial liquors are made with 6 and 3 samples, respectively. Given that classification models are calculated using large neural networks, these models are bound to be overfitted, especially because a proper validation of the classification models is impossible to implement. It is mentioned (lines 316-317) that data set was split in the training and testing data set, but in this case both data sets included measurements in the samples made on different days.

Author response: Dear reviewer, thanks a lot for your attentions. We are sorry that making your confusing on our study. We would like to make some explanations as follows:

  1. Thanks for your attentions to the number of samples in our experiments. In our study, our aim is to propose a machine learning method for the quality detection of base liquor and commercial liquor using multidimensional signals from an MOS-based E-nose. As we described in our manuscript, six types of base liquors and three types of commercial liquors were prepared for the experiments. The word “sample” has two understandings. The first only indicates the types of samples for classification (as you said, the base and commercial liquors are made with 6 and 3 types of samples, respectively). The second meaning is the number of sample data (here, the data is the measuring values from the E-nose). In our study, the experiments lasted for 25 days, and 54 base liquor samples (6 types of base liquors × 9 measurement times) and 27 commercial liquor samples (3 types of commercial liquors × 9 measurement times) were detected per day. Each sample was measured once and updated every day. And the experiments contained 2025 measurements (9 liquor samples × 9 measurement times × 25 day). Those numbers of data were conducted to training data set and test data set, respectively. In our study, the model for discrimination focuses on the number of sample data (the second meaning), not on the types of samples for classification (the first meaning). Therefore, the number of samples in our study is enough. According to your suggestions, we have revised the experiment description in Section 2.2. Instrument and Experiment (Line 121-125, Page 4, Clean revised version).

“…The experiments lasted 25 d; 54 base liquor samples (6 types of base liquors × 9 measurement times) and 27 commercial liquor samples (3 types of commercial liquors × 9 measurement times) were detected per day, and each sample was measured once and updated every day. Therefore, the experiments contained 2025 measurements (9 liquor samples × 9 measurement times × 25 d). ...”

  1. Thanks for your attention to the problem of overfitting for the classification models. As you said, overfitting is a fundamental issue in large neural networks which prevents us from perfectly generalizing the models to well fit observed data on training data, as well as unseen data on test set. Because of existence of overfitting, the model performs perfectly on training set, while fitting poorly on test set. This is due to that overfitted model has difficulty copying with pieces of the information in the test set, which may be different from those in the training set. The causes of this phenomenon might be the noise learning on the training set: when the training set is too small in size (measuring during a short period of time), or has fewer representative data (measuring the same samples repeatedly) [1]. Therefore, in order to reduce the effects of overfitting, we conducted the experiments for 25 days to obtain more data, and the liquor samples for daily experiments were updated to obtain more representative data (avoid the problem of overfitting as possibly in the process of data acquisition in the experiment stage). The data in the first 20 days was used for training and that in the last 5 days was used for test. So, the training dataset and test dataset are independent, which can used as a proper validation for the assessment of classification models.

For the design of the experiments, we have some introduction as follows:

  1. As is introduced in the literature [2], measurement uncertainty is one of the information losses connected with gas sensors, and gas sensor drift has a rather complex and inevitable effect for the sensor’s measurement properties. The time drift of gas sensor consists of a random temporal variation of the sensor response when it is exposed to the same analytes under identical conditions. The drift could make initial traditional machine learning method unsatisfactory for gas recognition after a relatively short period (typically few weeks) [3]. Frequent recalibrations are needed for to preserve accuracy. Therefore, in order to verify the robustness of our proposed method which could overcome the negative effect of time drift (to an extent), we conducted a set of experiments in 25 days. The data in the first 20 days was used for training and that in the last 5 days was used for testing. The experimental results indicated that our proposed model presented good performances and strong robustness for the classification of the nine Chinese liquors, which shows a potential to avoiding the negative effect of the time drift in E-nose.
  2. As far as we known, there are two principal approaches of building machine learning models: the model-driven (parametric) approach and the data-driven (non-parametric) approach [4]. The model-driven approach needs to design selectors (strategy or model to select samples from candidate data set) with parameter by handcraft feature or metric. For the data-driven approach, their selectors adopted deep architecture whose features are automatically generated but not by handcraft [5]. Therefore, the data-driven model’s performance can be significantly affected by the quantity and quality of dataset. Model training is actually a process of tuning its hyper-parameter. Well-tuned parameters make a good balance between training accuracy and regularity, and then inhibit the effect of overfitting. To tune these parameters, the model needs sufficient samples for learning. Therefore, in order to reduce the effects of overfitting, we chose to conduct the experiments for 25 days to obtain more data. The liquor samples for daily experiments were updated to obtain more representative data. The data in the first 20 days was used for training and that in the last 5 days was used for testing. The experimental results indicated that our proposed model achieved a good performance on both the training and test set.
  3. Generally, a series of measurements conducted under the same conditions is called an equal-precision measurement [6]. Strictly speaking, there is no equal-precision measurement in the practical applications of E-nose. In our study, the experiments were conducted for 25 days in a clean testing room of the author’ laboratory (with good ventilation and an area of about 45 square meters) at a temperature of 26 °C ± 2 °C and a humidity level of 50% ± 2%. All the experiments were performed by a professional and experienced operator using the same devices. So, we think that there is no gross error in our measurements. The nine Chinese liquors (six base liquors and three commercial liquors) were measured nine times per day and the liquor samples for daily experiments were updated. So, the measurements conducted on different days were independent and could be regarded as unequal-precision measurements. These independent measurements satisfy the conditions of E-nose applications in real scenarios and improve the reliability and authenticity of the dataset, which can be used to verify the generalization of proposed method. In addition, as we know, the calculation of fusion weight is essential in the unequal-precision measurement, which limits the data process precision by traditional statistical method [7]. Our proposed method does not have to calculate the weights, and has good performances for the classification of Chinese liquors.
  4. In this study, we hope to propose a data-driven model to classify base liquor and commercial liquor using multidimensional signals from an MOS-based E-nose. In odor to solve the problem of overfitting, a LightGBM classifier was employed by us. The advantages of the LightGBM classifier are as follows:
  5. LightGBM uses a histogram algorithm to replace the pre-sorted algorithm of XGBoost. This algorithm discretizes continuous feature values into K integers and constructs a histogram with a width of K. When traversing, the histogram algorithm takes the discretized value as index and accumulates the statistics in the histogram. After one traversal, an optimal segmentation point can be found according to the histogram. In fact, due to the discretization of features, the segmentation points are not accurate. However, decision tree is a weak learner. Adopting the histogram algorithm can regularize classification model and prevent overfitting. Then, the histogram algorithm improves training efficiency and reduces the demand for memory space [8].
  6. LightGBM uses Leaf-wise strategy to replace the level-wise strategy of decision tree. This strategy is a more effective strategy, which finds the leaves with the highest branching gain each time from all the leaves, and then goes through the branching cycle. Therefore, compared with the horizontal direction, the blade can reduce more errors and obtain better precision with the same number of times of segmentation. The downside of leaf orientation is that it can grow deeper decision trees and produce overfitting. Therefore, LightGBM adds a maximum depth limit to the top of the leaf to prevent overfitting while ensuring high efficiency [9].

Thus, a ResNet-GBM framework consisting of a ResNet18 backbone and a LightGBM classifier was proposed in this study. In the framework, the ResNet18 backbone is a powerful feature extractor and automatically extracts a sufficient number of comprehensive and significant features from the raw multidimensional E-nose signals. The LightGBM is employed to strengthen the identification ability of the liquor’s quality. The effectiveness of this framework was verified by comparing with other six machine learning models (SVM, RF, KNN, XGBoost, MDS-SVM, and BPNN) with the best performance on five evaluation metrics (accuracy, sensitivity, precision, F1-score, and Kappa score). The results showed that the proposed model has strong potential for the quality detection of base liquor and commercial liquor.

  1. Thanks for your attention to the training data set and test data set in our study. As you said, the training data set and test data set included measurements in the samples made on different days. However, the two data sets are not split by ourselves and are totally independent. The nine Chinese liquors (six base liquors and three commercial liquors) were measured nine times under the same conditions per day and the liquor samples for daily experiments were updated. These independent measurements satisfy the conditions of E-nose applications in real scenarios and improve the reliability and authenticity of the dataset, which can be used to verify the robustness and generalization of our proposed method.

References:

[1] Ying, X. (2019, February). An overview of overfitting and its solutions. In Journal of Physics: Conference Series (Vol. 1168, No. 2, p. 022022). IOP Publishing.

[2] Boeker, P. (2014). On ‘electronic nose’methodology. Sensors and Actuators B: Chemical, 204, 2-17.

[3] Padilla, M., Perera, A., Montoliu, I., Chaudry, A., Persaud, K., & Marco, S. (2010). Drift compensation of gas sensor array data by orthogonal signal correction. Chemometrics and Intelligent Laboratory Systems, 100(1), 28-35.

[4] Tarsha-Kurdi, F., Landes, T., Grussenmeyer, P., & Koehl, M. (2007, September). Model-driven and data-driven approaches using LIDAR data: Analysis and comparison. In ISPRS workshop, photogrammetric image analysis (PIA07) (pp. 87-92).

[5] Liu, P., He, G., & Zhao, L. (2021). From Model-driven to Data-driven: A Survey on Active Deep Learning. arXiv preprint arXiv:2101.09933.

[6] Zheng, H., & Gu, Y. (2021). EnCNN-UPMWS: Waste Classification by a CNN Ensemble Using the UPM Weighting Strategy. Electronics, 10(4), 427.

[7] Wang, J., He, Z., Zhou, H., Li, S., & Zhou, X. (2017). Optimal weight and parameter estimation of multi-structure and unequal-precision data fusion. Chinese Journal of Electronics, 26(6), 1245-1253.

[8] Zhang Y, Zhao X, Li Z. Facilitated and Enhanced Human Activity Recognition via Semi-supervised LightGBM[C]//2020 IEEE Globecom Workshops (GC Wkshps. IEEE, 2020: 1-6.

[9] Fan J, Ma X, Wu L, et al. Light Gradient Boosting Machine: An efficient soft computing model for estimating daily reference evapotranspiration with local and external meteorological data[J]. Agricultural water management, 2019, 225: 105758.

Reviewer#1, Concern # 2:

- PCA score plot presented in the Fig. 5 does not provide assurance that e-nose is capable to distinguish liquor samples as they overlap. What PCA score plot does show is that there is significant drift in the data. Considering this, the data sets collected in this work could be more suitable for the study of drift reduction than to develop classification models.

Author response: Dear reviewer, thanks a lot for your attentions and suggestions.

  1. In this study, our main aim is proposing a machine learning method for the quality detection of base liquor and commercial liquor by means of the MOS-based E-nose. PCA, an unsupervised method commonly used for pattern recognition, was first used to distinguish the nine liquor samples as a regular tool. But the results of the PCA was very poor as you said. The PCA score plot presented in the Figure 5 (Figure 5. Projection of the first three principal components of the PCA of the liquor samples, Page 9, Clean revised version) showed a relatively high overlap between the different liquor samples. The poor PCA results on distinguishments indicated that the differences among the samples were relatively tiny. Some meaningful feature information for the classification was lost during dimensionality reduction. As you said, the PCA is not capable to distinguish liquor samples by means of the E-nose in our case. Therefore, we proposed a novel framework, namely ResNet-GBM, to mine the deep aroma features of the liquors for the quality detection of the nine liquor samples.
  2. As you said, there is significant drift in the data according to the PCA score plot. As we answered the concern #1, the time drift of gas sensor consists of a random temporal variation of the sensor response when it is exposed to the same analytes under identical conditions. Therefore, we chose to conducted the experiments in 25 days. So, the experiment data satisfy the conditions of E-nose applications in real scenarios and improve the reliability and authenticity of the dataset. The experimental results indicated that our proposed model presented good performances and strong robustness for the classification of the nine Chinese liquors, which shows a potential to avoiding the negative effect of the time drift in E-nose. In addition, we are so grateful that you provided us a new research direction. We would like to make further research on the questions of data drift for E-nose in the future.

Thanks again for your professional suggestions.

Reviewer#1, Concern # 3:

- Please provide a table with analyzed base liquor listed, their age, and alcohol content of all base and commercial liquors analyzed.

Author response: Dear reviewer, thanks a lot for your attentions and suggestions.

According to your suggestions, we have added the details of base and commercial liquors you mentioned in Table 1 (Table 1, Page 3, Clean revised version) and Table 2 (Table 2, Page 3, Clean revised version). Now the details of samples are presented more clearly. Thanks so much. And we hope the revision could meet your requirements.

“…Table 1. Details of the base liquor.

Label

Aging duration

Alcohol content

BL13

13

65

BL11

11

65

BL8

8

65

BL6

6

65

BL5

5

65

BL3

3

65

…Table 2. Details of the commercial liquors.

Label

Production year

Alcohol content

Price($/450ml)

Components

Blending proportion (%)

CL1

2022

42

67.2

BL13

0.01

BL11

8.99

BL5

31

BL3

60

CL2

2022

42

49.1

BL13

0.008

BL8

5.992

BL5

24

BL3

70

CL3

2022

42

37.7

BL13

0.006

BL6

3.994

BL5

16

BL3

80

…”

Reviewer#1, Concern # 4:

- Ethanol is a major compound of liquors, and it can influence sensor response. Have sensors’ sensitivity to ethanol been studied?

Author response: Dear reviewer, thanks a lot for your attentions.

In this study, a well-known commercial E-nose (PEN3, produced by Airsense Analytics GmbH, Germany) was used to acquire the characteristic flavor information of the liquor samples. The PEN3 is based on a metal-oxide gas sensor array which consisted of 10 MOS-based sensors. As shown in Table 3 (Table 3, Page 3, Clean revised version), we have listed the main performance of the 10 sensors in the PEN3 E-nose. Each sensor in the PEN3 E-nose is typically not sensitive for a single kind of substance, but rather for a certain range of substance (more details can be found on the website of the PEN3, www.airsense.com). As far as we known, the PEN3 E-nose has been used to distinguished different alcoholic beverages [1-2]. In our pervious study, we have employed the PEN3 E-nose to distinguish wine and Chinese liquors by means of a machine learning technique successfully [3].

Chinese liquor is a popular alcoholic beverage which is produced from grains using traditional methods, including fermentation, distillation, storage, and blending. Among the complex manufacturing process of the Chinese liquor, many flavoring substances are produced, including alcohols, esters, aldehydes, ketones, phenols, acids, nitrogen compounds and sulfides [4]. As we introduced above, the PEN3 E-nose was employed to acquire the comprehensive information of volatile compound from the liquor samples in our study. Although ethanol is a major compound of the liquors, what we focused on is the deep aroma features of the liquors which is critical for the classification of different liquor samples. Therefore, we proposed a machine learning method, namely ResNet-GBM, to automatically mine the deep aroma features using multidimensional signals from the E-nose.

References:

[1] Xia, Y., Zha, M., Liu, H., Shuang, Q., Chen, Y., & Yang, X. (2022). Novel Insight into the Formation of Odour—Active Compounds in Sea Buckthorn Wine and Distilled Liquor Based on GC–MS and E–Nose Analysis. Foods, 11(20), 3273.

[2] Macías, M. M., Manso, A. G., Orellana, C. J. G., Velasco, H. M. G., Caballero, R. G., & Chamizo, J. C. P. (2012). Acetic acid detection threshold in synthetic wine samples of a portable electronic nose. Sensors, 13(1), 208-220.

[3] Yang, Y., Liu, H., & Gu, Y. (2020). A model transfer learning framework with back-propagation neural network for wine and Chinese liquor detection by electronic nose. IEEE Access, 8, 105278-105285.

[4] Hong, J., Tian, W., & Zhao, D. (2020). Research progress of trace components in sesame-aroma type of baijiu. Food Research International, 137, 109695.

Reviewer#1, Concern # 5:

- Chinese liquor composition has been extensively studied. Please provide some overview of the main compounds that are believed to contribute to the liquor quality.

Author response: Dear reviewer, thanks a lot for your attentions and suggestions.

According to your suggestions, we have added some overview of the main compounds that are believed to contribute to the liquor quality in Section 1. Introduction (Line 41-51, Page 2, Clean revised version). Now the readability of our revised manuscript is increased. Thanks so much. And we hope the revision could meet your requirements.

“…The main components of Chinese liquors are alcohol and water, accounting for 98% of the total weight, and there is usually less than 2% of other trace components but contributes to the complex aroma of the liquors, including esters, aldehydes, ketones, phenols, acids, nitrogen compounds and sulfides [1]. The most common method for assessing the quality of base liquors is sensory evaluation and chemical/spectroscopic analysis [8] ...”

Reviewer#1, Concern # 6:

- What was the rational for sensor selection for an e-nose? For examples, sensors sensitive primarily to hydrogen (W6S) and alkanes (W3S) were included in the sensor array, though obviously these compounds are not expected to be present in the liquor headspace.

Author response: Dear reviewer, thanks a lot for your attentions.

As we answer for the concern #4, in our study, we used a well-known commercial E-nose (PEN3, produced by Airsense Analytics GmbH, Germany) with 10-channel MOS-based sensors to acquire the comprehensive information of volatile compound from the liquor samples. As far as we known, the PEN3 E-nose has been widely applied in the study of distinguishing different alcoholic beverages [1-2]. In our pervious study, we have employed the PEN3 E-nose to distinguish wine and Chinese liquors by means of a machine learning technique successfully [3].

Although some sensors sensitive primarily to hydrogen (W6S) and alkanes (W3S) are included in the sensor array and these compounds are not expected to be present in the liquor headspace as you said, what we focused on is the comprehensive information from the multidimensional signals of the E-nose. That information is automatically mined and analyzed by means of the proposed ResNet-GBM framework. The results of our study showed an excellent performance of our proposed framework for the quality detection of base liquor and commercial liquor using multidimensional signals from an E-nose.

References:

[1] Xia, Y., Zha, M., Liu, H., Shuang, Q., Chen, Y., & Yang, X. (2022). Novel Insight into the Formation of Odour—Active Compounds in Sea Buckthorn Wine and Distilled Liquor Based on GC–MS and E–Nose Analysis. Foods, 11(20), 3273.

[2] Macías, M. M., Manso, A. G., Orellana, C. J. G., Velasco, H. M. G., Caballero, R. G., & Chamizo, J. C. P. (2012). Acetic acid detection threshold in synthetic wine samples of a portable electronic nose. Sensors, 13(1), 208-220.

[3] Yang, Y., Liu, H., & Gu, Y. (2020). A model transfer learning framework with back-propagation neural network for wine and Chinese liquor detection by electronic nose. IEEE Access, 8, 105278-105285.

Reviewer#1, Concern # 7:

- Table 3 is unnecessary, explanation given in the text is sufficient.

Author response: Dear reviewer, thanks a lot for your attentions and suggestions.

According to your suggestions, we have removed the Table 3 in our manuscript and revised the description in Section 2.7. Model Evaluation Metrics (Line 205-207, Page 6, Clean revised version). Now the readability of our revised manuscript is increased. Thanks again.

“…Model evaluation metrics were employed to assess the performance of the supervised learning algorithm. Four parameters were used to calculate the metrics: True Positive (TP), False Positive (FP), True Negative (TN), and False Negative (FN) ...”

Reviewer#1, Concern # 8:

- Recall and precision are terms used for sensitivity and specificity in machine learning. As the Foods is a food science and technology journal, I would suggest to use terminology with which its readers are more familiar, i.e. sensitivity and specificity.

Author response: Dear reviewer, thanks a lot for your attentions and suggestions.

Thanks a lot for pointing out the better terminology here. Recall (sensitivity) is defined as the ratio of the True Positive (TP) samples to the sum of the TP and False Negative (FN) samples, which indicate the true positive rate. Precision is defined as the ratio of the TP samples to the sum of the TP and False Positive (FP) samples, which indicates the positive predictive value. Specificity is defined as the ratio of the True Negative (TN) samples to the sum of the FP and TN samples, which indicates the true negative rate.

In odor to verify the performance of the proposed method, five evaluation metrics are used in our study, including accuracy, recall (sensitivity), precision, F1 score, and kappa score. According to your suggestions, we have replaced the “recall” with the “sensitivity” and remained the “precision” in our manuscript.

The readability of our revised manuscript is increased by means of your suggestions. Thanks so much. And we hope the revision could meet your requirements.

Reviewer#1, Concern # 9:

- Classification results presented in the tables 5-9 refer to train or test data set?

Author response: Dear reviewer, thanks a lot for your attentions.

As we described in Section 3. Proposed Method (Line 257-258, Page 8, Clean revised version), the classification results presented in the Table 5-9 refer to test data set. As far as we known, the classification results on the training data set represents the training results of the models, which is an intermediate process. In our study, we presented the final results of the models on the test data set.

We hope our responses can remove your concerns and the revised version can meet your requirements.

Reviewer 2 Report

Dear Authors,

The current manuscript reports the machine learning method for the quality detection of base liquor and commercial liquor using multidimensional signals from an electronic nose. In general, this is an important and interesting work. The manuscript is original. There are very few works aimed at such a deep research of the evaluating process of the product quality using an electronic nose. The manuscript is presented in a competent, scientific language, logically structured, and easy to read. The conclusions are consistent with the results obtained and correspond to the goal of the study.

I have a few remarks and comments.

Add the abbreviation CL to the text before Table 1.

Make a link to the regulatory documentation, in accordance with which the conditions for the experiment were selected.

In general, the manuscript describes in detail the methodology of the experiment, the results of the experiment, but in the Results and Discussion there are no discussions with the results of other authors.

Author Response

Original Manuscript ID: foods-2204718     

Original Article Title: “A Machine Learning Method for the Quality Detection of Base Liquor and Commercial Liquor Using Multidimensional Signals from an Electronic Nose

Reviewer #2 (Comments and Suggestions for Authors):
Dear Authors,

The current manuscript reports the machine learning method for the quality detection of base liquor and commercial liquor using multidimensional signals from an electronic nose. In general, this is an important and interesting work. The manuscript is original. There are very few works aimed at such a deep research of the evaluating process of the product quality using an electronic nose. The manuscript is presented in a competent, scientific language, logically structured, and easy to read. The conclusions are consistent with the results obtained and correspond to the goal of the study.

I have a few remarks and comments.

Add the abbreviation CL to the text before Table 1.

Make a link to the regulatory documentation, in accordance with which the conditions for the experiment were selected.

In general, the manuscript describes in detail the methodology of the experiment, the results of the experiment, but in the Results and Discussion there are no discussions with the results of other authors.

Dear reviewer,

On the arrival of the Chinese Spring Festival, extend you all my best wishes for your perfect health and lasting prosperity.

Thanks for your supporting. Our answers for your questions are as follows:

Reviewer #2, Concern # 1:

- Add the abbreviation CL to the text before Table 1.

Author response: Dear reviewer, thanks a lot for your attentions and suggestions.

According to your suggestions, we have added the abbreviation CL to the text before Table 2 in Section 2.1 Chinese liquor samples (Line 109, Page 3, Clean revised version). Now the readability of our revised manuscript is increased. Thanks so much. And we hope the revision could meet your requirements.

“…The three commercial liquors (CL) were blended using different proportions of the six base liquors. The details are listed in Table 2 …”

Reviewer#2, Concern # 2:

- Make a link to the regulatory documentation, in accordance with which the conditions for the experiment were selected.

Author response: Dear reviewer, thanks a lot for your attentions.

In this study, a well-known commercial E-nose (PEN3, produced by Airsense Analytics GmbH, Germany) was used to acquire the volatile compound profile of the liquor samples (more details can be found on the website of the PEN3, www.airsense.com). As a powerful odor analysis device, the PEN3 E-nose has been used to distinguish different alcoholic beverages [1-2]. In our pervious study, we have employed the PEN3 E-nose to distinguish wine and Chinese liquors by means of a machine learning technique successfully [3]. Therefore, in this study, we conducted the experiments under the same conditions (temperature: 26 ± 2 °C; relative humidity: 50 ± 2%) in a single clean testing room of our laboratory with good ventilation. And the workflow of the E-Nose (including the collection stage and flushing stage) was strictly performed according to the handbook of the PEN3 E-nose which is provided by the manufacturers. Therefore, we think that the experiment conditions in our study is reasonable and reliable. According to your suggestions, we have added the link of the handbook in Section 2.2 Instrument and Experiment (Line 147-148, Page 5, Clean revised version).

“…The workflow of the E-nose includes the collection stage and flushing stage. Before the measurement, clean air was pumped through filter 2 into the E-nose with a flow rate of 10 mL/s for 100 s. The automatic adjustment and calibration of the zero gas are called the zero-point trim; the values relative to the zero-point values were recorded as a baseline. After the calibration, the liquor sample’s volatile gas in the sampler was pumped into the E-nose with a flow rate of 10 mL/s to contact the sensor array for 100 s. The gas molecules were adsorbed on the sensors’ surface, changing the sensors’ conductivity due to the redox reaction on the surface of the sensor’s active element. The sensors’ conductivity eventually stabilized at a constant value when the adsorption was saturated. The collection stage lasted 100 s, and sampling continued at one sample per second. The sensor chamber was flushed with zero gas between measurements to remove the analytes. The flushing and data collection stages were repeated to obtain the raw data of the nine liquor samples. All the workflow of the E-Nose was strictly performed according to the handbook of the PEN3 E-nose which is provided by the manufacturers [19] …”

Thanks again for your attentions and suggestions. And we hope our responses can remove your concerns.

References:

[1] Xia, Y., Zha, M., Liu, H., Shuang, Q., Chen, Y., & Yang, X. (2022). Novel Insight into the Formation of Odour—Active Compounds in Sea Buckthorn Wine and Distilled Liquor Based on GC–MS and E–Nose Analysis. Foods, 11(20), 3273.

[2] Macías, M. M., Manso, A. G., Orellana, C. J. G., Velasco, H. M. G., Caballero, R. G., & Chamizo, J. C. P. (2012). Acetic acid detection threshold in synthetic wine samples of a portable electronic nose. Sensors, 13(1), 208-220.

[3] Yang, Y., Liu, H., & Gu, Y. (2020). A model transfer learning framework with back-propagation neural network for wine and Chinese liquor detection by electronic nose. IEEE Access, 8, 105278-105285.

Reviewer#2, Concern # 3:

- In general, the manuscript describes in detail the methodology of the experiment, the results of the experiment, but in the Results and Discussion there are no discussions with the results of other authors.

Author response: Dear reviewer, thanks a lot for your attentions and suggestions.

According to your suggestions, we have added the discussions with the results of two other authors in the Results and Discussion in Section 4. Results and Discussion (Line 278-363, Page 9-12, Clean revised version). Now the readability of our revised manuscript is increased. Thanks so much. And we hope the revision could meet your requirements.

“…Experiments II to IV were performed to compare the proposed ResNet-GBM framework with six other methods (including four common machine learning methods (SVM, RF, KNN, and XGBoost) and two methods proposed by other authors (MDS-SVM and BPNN)) on Dataset A, Dataset B, and Dataset C, respectively. MDS-SVM is a pattern recognition method based on multidimensional scaling and SVM which was developed to classify ten brands of Chinese liquors by Li et al [28]. BPNN is a multi-layered feedforward neural network which was used to distinguish different wines based on their properties by Liu et al [29] ...

…The classification results derived from the five seven models on the test set are displayed in Table 6. Five evaluation metrics were used to evaluate the classification models. As shown in Table 6, the proposed ResNet-GBM obtained the best performance for all evaluation metrics, with accuracy of 0.9704, sensitivity of 0.9704, precision of 0.9716, F1 score of 0.9710, and kappa score of 0.9644. SVM, RF, KNN, XGBoost, MDS-SVM and BPNN achieved accuracies of 0.3175, 0.4018, 0.4053, 0.4246, 0.7852 and 0.8963, respectively. Obviously, the performances of these six models were unsatisfactory because they failed to extract a sufficient number of deep features. The experimental results demonstrated the effectiveness and superior performance of the proposed ResNet-GBM framework to identify different base liquors…

Table 6. Results of five evaluation metrics for seven models in Experiment II.

Model

Accuracy

Sensitivity

Precision

F1 score

Kappa score

SVM

0.3175

0.3175

0.3531

0.3042

0.1811

RF

0.4018

0.4018

0.3696

0.3590

0.2821

KNN

0.4053

0.4053

0.3732

0.3768

0.2863

XGBoost

0.4246

0.4246

0.4437

0.4096

0.3095

MDS-SVM [28]

0.7852

0.7826

0.8206

0.7740

0.7419

BPNN [29]

0.8963

0.8962

0.8995

0.8948

0.8755

ResNet-GBM

0.9704

0.9704

0.9716

0.9710

0.9644

…The parameters of the models were the same as those in Experiment II. As shown in Table 7, the ResNet-GBM model exhibited the best results for the commercial liquors, with accuracy of 0.9814, sensitivity of 0.9814, precision of 0.9825, F1 score of 0.9815, and kappa score of 0.9722. SVM, RF, KNN, XGBoost, MDS-SVM and BPNN achieved accuracies of 0.3649, 0.6351, 0.5088, 0.6105, 0.8235 and 0.9118, respectively. The results showed that the proposed model could accurately detect different grades of commercial liquor. The classification performance of the model was better for the commercial liquor than for the base liquor.

Table 7. Results of five evaluation metrics for seven models in Experiment III.

Model

Accuracy

Sensitivity

Precision

F1 score

Kappa score

SVM

0.3649

0.3649

0.3677

0.3501

0.2379

RF

0.6351

0.6351

0.6453

0.6322

0.5237

KNN

0.5088

0.5088

0.5099

0.5086

0.4912

XGBoost

0.6105

0.6105

0.6767

0.6165

0.5064

MDS-SVM [28]

0.8235

0.8202

0.8737

0.8206

0.7345

BPNN [29]

0.9118

0.9130

0.9183

0.9115

0.8677

ResNet-GBM

0.9814

0.9814

0.9825

0.9815

0.9722

…The results of the seven models are listed in Table 8. The proposed ResNet-GBM model achieved the best performance with accuracy of 0.9803, sensitivity of 0.9803, precision of 0.9819, F1 score of 0.9801, and kappa score of 0.9778 for the simultaneous classification of the base liquors and commercial liquors. SVM, RF, KNN, XGBoost, MDS-SVM and BPNN achieved accuracies of 0.3389, 0.4468, 0.4304, 0.4901, 0.8148 and 0.9074, respectively. The comparison results indicate that the proposed method provided better performances for mining deep features from the sensor signals and demonstrated that the proposed ResNet-GBM model provided superior performance for the classification of base liquors, commercial liquors, and a mixture of both.

Table 8. Results of five evaluation metrics for seven models in Experiment IV.

Model

Accuracy

Sensitivity

Precision

F1 score

Kappa score

SVM

0.3389

0.3389

0.3299

0.3268

0.2563

RF

0.4468

0.4468

0.4764

0.4290

0.3776

KNN

0.4304

0.4304

0.4141

0.4139

0.3592

XGBoost

0.4901

0.4901

0.5337

0.4849

0.4263

MDS-SVM [28]

0.8148

0.8148

0.8420

0.8161

0.7917

BPNN [29]

0.9074

0.9074

0.9170

0.9074

0.8958

ResNet-GBM

0.9803

0.9803

0.9819

0.9801

0.9778

…”

Reviewer 3 Report

Te content of this manuscript is of great interest for the readers.

Minor English language style improvement is needed.

Lines 42-45 - I would not address the exact price especially in these times.

Please check the lines 46-50 as they are confusing. Please rephrase or explain how can the health condition, emotional states, etc., influence the chemical/spectroscopic analysis.

Table 1 - exaplain the importance of specifying the price here.

Do not use list numbering for the Conclusions section.

Author Response

Original Manuscript ID: foods-2204718     

Original Article Title: “A Machine Learning Method for the Quality Detection of Base Liquor and Commercial Liquor Using Multidimensional Signals from an Electronic Nose

Reviewer #3 (Comments and Suggestions for Authors):
Te content of this manuscript is of great interest for the readers.

Minor English language style improvement is needed.

Lines 42-45 - I would not address the exact price especially in these times.

Please check the lines 46-50 as they are confusing. Please rephrase or explain how can the health condition, emotional states, etc., influence the chemical/spectroscopic analysis.

Table 1 - exaplain the importance of specifying the price here.

Do not use list numbering for the Conclusions section.

Dear reviewer,

On the arrival of the Chinese Spring Festival, extend you all my best wishes for your perfect health and lasting prosperity.

Thanks for your supporting. Our answers for your questions are as follows:

Reviewer #3, Concern # 1:

- Lines 42-45 - I would not address the exact price especially in these times.

Author response: Dear reviewer, thanks a lot for your attentions and suggestions.

According to your suggestions, we have removed the exact price in Section 1. Introduction (Line 42-45, Page 1-2, Clean revised version). Now the readability of our revised manuscript is increased. Thanks so much. And we hope the revision could meet your requirements.

“…For example, Maotai is the most famous brand of Chinese liquor and is used to create many popular products. However, due to the different base liquor qualities, the price of Maotai Feitian liquor is dozens of times higher than that of Maotai Prince liquor [7] …”

Reviewer#3, Concern # 2:

- Please check the lines 46-50 as they are confusing. Please rephrase or explain how can the health condition, emotional states, etc., influence the chemical/spectroscopic analysis.

Author response: Dear reviewer, thanks a lot for your attentions.

We are so sorry that making your confusing because of the unclear descriptions. We would like to make some explanations. As far as we known, the most common methods for assessing the quality of base liquor are sensory evaluation and chemical/spectroscopic analysis. However, for the sensory evaluation method, the accuracy and objectivity of the results cannot be guaranteed because experts may be influenced by their health conditions, emotional states, or environmental factors. Analysis methods, such as chromatography and spectroscopy, are demanding and time-consuming. Therefore, the aim of our study is to develop an objective, convenient, rapid, and accurate method to detect the quality detection of base liquor.

We have revised the descriptions in Section 1. Introduction (Line 50-55, Page 2, Clean revised version). Now the rigor and readability of our revised manuscript is increased. Thanks so much. And we hope the revision could meet your requirements.

“…The most common method for assessing the quality of base liquors is sensory evaluation and chemical/spectroscopic analysis [8]. However, for the sensory evaluation method, the accuracy and objectivity of the results cannot be guaranteed because experts may be influenced by their health conditions, emotional states, or environmental factors [9]. Analysis methods, such as chromatography [10] and spectroscopy [11], are demanding and time-consuming...”

Reviewer#3, Concern # 3:

- Table 1 - exaplain the importance of specifying the price here.

Author response: Dear reviewer, thanks a lot for your attentions.

In this study, our main aim is proposing a machine learning method for the quality detection of base liquor and commercial liquor using multidimensional signals form an MOS-based E-nose. As we known, price is an important representation for the quality grades of commercial liquors. Therefore, we remained the price information of the commercial liquor in the revised Table 2 (Table 2, Page 3, Clean revised version) to present that even the commercial liquors in the same brand have different quality grades. We hope our responses can remove your concerns. Thanks again.

Reviewer#3, Concern # 4:

- Do not use list numbering for the Conclusions section.

Author response: Dear reviewer, thanks a lot for your attentions and suggestions.

According to your suggestions, we have removed the list numbering for the Conclusions section in Section 5. Conclusions (Line 378-411, Page 12-13, Clean revised version). Now the rigor and readability of our revised manuscript is increased. Thanks so much. And we hope the revision could meet your requirements.

“…In this study, a ResNet-GBM framework was proposed to identify base liquors and commercial liquors with different qualities using a MOS-based E-nose. The main conclusions are as follows:

PCA was used to distinguish nine liquors samples using the E-nose data. Highly coincident points in the PCA result indicated that the odor information of different liquors was highly similar. The PCA results showed that this method was ineffective for distinguishing liquors with different qualities, and meaningful feature information was lost during dimensionality reduction.

A ResNet-GBM framework consisting of the ResNet18 backbone and the LightGBM classifier was proposed to perform the classification. Ablation experiments were conducted to determine the contributions of the ResNet-GBM’s components for identification. The results indicated the effectiveness of the proposed framework. The significant features contained in the multidimensional signals were extracted by the ResNet18 backbone. The LightGBM classifier improved the identification ability of the ResNet model, and the proposed model achieved classification accuracies of 0.9704, 0.9814, and 0.9803 for Datasets A, B, and C, respectively.

The superiority of the proposed framework was demonstrated by comparing it with six other methods (SVM, RF, KNN, XGBoost, MDS-SVM, and BPNN) using the three datasets. The comparative experiments proved that the proposed framework had higher classification performance and better generalization ability than the other models using the multidimensional E-nose signals as input.

The F1 scores of the ResNet-GBM model for all samples were compared using the three datasets (base liquor dataset, commercial liquor dataset, and mixed dataset). The proposed ResNet-GBM model achieved better performance for the classification of commercial liquor using the mixed dataset (1.0000 for CL1, CL2, and CL3) than the commercial liquor dataset (0.9730 for CL1, 0.9714 for CL2, and 1.0000 for CL3). The results indicated that the excellent performance for distinguishing base liquors resulted in a higher classification accuracy of commercial liquors when base liquors and commercial liquors were analyzed simultaneously.

The results were encouraging and demonstrated that a deep learning framework could be used to identify base liquors and commercial liquors with different qualities using E-nose data. This approach provides a potential tool for the quality control of liquor and promotes the practical application of E-nose devices. We expect that this framework has broad application value for using deep learning methods for food quality control.”

Round 2

Reviewer 1 Report

The authors made some changes to the manuscript, however, the main issues of this work have not been addressed satisfactorily.

The main problem with this work is limited number of analysed samples. There is a basic misunderstanding on what constitutes an independent sample. Several measurements in the same sample, made on the same or on different days, are not independent samples. These are replicated measurements. Chemical composition of a sample remains exactly the same independently of how many times it was analyzed. 

If analytical instrument used for measurements was ideal, its response in the same sample would be exactly the same in all repeated measurements. But in reality, there are always differences in the instrumental response due to the drift, and in the case of e-nose, as PCA score plot demonstrates, these differences are quite significant, which could be expected as e-noses are prone to drift.

In this work classification model was calculated using 1 sample per class, which is clearly inadequate. If another liquor sample belonging to e.g. the class BL13 is presented to such classification model, it would be misclassified as there are slight differences in composition between liquor samples of the same type.

The large data set acquired in this work could be used for study of the drift reduction / recalibration approaches, which would require reformulation of the machine learning method. Alternatively, the authors can apply proposed classification approach to the other data set.

Author Response

Original Manuscript ID: foods-2204718     

Original Article Title: “A Machine Learning Method for the Quality Detection of Base Liquor and Commercial Liquor Using Multidimensional Signals from an Electronic Nose

Reviewer #1 (Comments and Suggestions for Authors):
The authors made some changes to the manuscript, however, the main issues of this work have not been addressed satisfactorily.

The main problem with this work is limited number of analysed samples. There is a basic misunderstanding on what constitutes an independent sample. Several measurements in the same sample, made on the same or on different days, are not independent samples. These are replicated measurements. Chemical composition of a sample remains exactly the same independently of how many times it was analyzed.

If analytical instrument used for measurements was ideal, its response in the same sample would be exactly the same in all repeated measurements. But in reality, there are always differences in the instrumental response due to the drift, and in the case of e-nose, as PCA score plot demonstrates, these differences are quite significant, which could be expected as e-noses are prone to drift.

In this work classification model was calculated using 1 sample per class, which is clearly inadequate. If another liquor sample belonging to e.g. the class BL13 is presented to such classification model, it would be misclassified as there are slight differences in composition between liquor samples of the same type.

The large data set acquired in this work could be used for study of the drift reduction / recalibration approaches, which would require reformulation of the machine learning method. Alternatively, the authors can apply proposed classification approach to the other data set.

Author response: Dear reviewer, thanks a lot for your attentions and suggestions.

We totally agree with your opinion about the constitution of independent sample. Actually, in our study, the design of our experiments was same with your minds that all the measurements were absolutely on the independent samples. However, we did not clearly describe the details of the experiment samples in our manuscript, which caused your misunderstanding. We are also sorry that we did not answer the question of the samples in our study so clearly in response (Round 1). Now, our explanations are as follows:

  1. As you said, several measurements in the same sample, made on the same or on different days, are not independent samples. As we introduced in our manuscript, in our study, all the liquor samples were provided by the Shanxi Luxian Liquor Industry Co., Ltd. The experiments lasted for 25 days, and six types of base liquors and three types of commercial liquors were prepared for the experiments (here, we point out that nine types of liquor samples). As regards each type of liquors, nine different individual samples (the base liquors were not from the same liquor storage, and the commercial liquors were not from the same bottle) with same volume (10ml) were provided everyday by the manufacturer and measured using E-nose (total 25 days × 9 individual samples = 225 individual samples per type). Therefore, the total number of samples in our study was 2025 (9 types of liquor samples × 225 individual samples per type). Each sample was measured once and updated if it was used, which ensures that no repeated measurements exist in the experiments. Therefore, the measurements on the same or on different days were not in the same samples (replicated measurements), but in the independent samples (independent measurements). According to your suggestions, we have revised the experiment description in Section 2.2. Instrument and Experiment (Line120-129, Page 4, Clean revised version).

“…The experiments lasted for 25 days, and nine different individual samples of each type were measured every day within the same procedure (total 25 days × 9 individual samples = 225 individual samples per type). These individual samples were from different production batches (the base liquor samples were from different liquor storages, and the commercial liquor samples were from different bottles) and provided by the manufacturer directly. Each sample was measured once and updated if it was used, which ensured that no repeated measurements existed in the experiments. Therefore, the experiments contained 2025 independent measurements (9 types of liquor samples × 225 individual samples per type) ….”

  1. In our study, our aim is to propose a machine learning method for the quality detection of base liquor and commercial liquor using multidimensional signals from an MOS-based E-nose. As far as we known, in the researches of food quality detection by means of an E-nose system, almost all the researchers conducted the experiments using samples from different batches and some researchers even conducted repeat measurements on the samples from the same batch. Zhang et al. [1] proposed a channel attention convolutional neural network for the detection of 10 kinds of Chinese liquor, with each liquor being presented in six different production batches. As regards each type of liquors in different batches, 20 samples with 50ul were obtained by reusing the dropper (total 6 batches × 20 measurement times = 120 samples per kind). Li et al. [2] used an E-nose to recognize odors and diagnose quality of bacon. Five different batches of manufactured bacon were sampled and for each production batch 18 parallel samples were selected randomly, a total of 90 samples of smoked bacon were tested in experiments (total 5 batches × 18 measurement times = 90 samples per kind). Xu et al. [3] proposed a rapid detection method of jointly using E-nose and computer vision system to detect tea aroma and tea appearance for tea quality identification. And 32 replicates for each tea grade were detected by the E-nose (32 samples per kind). Therefore, we think that the samples (total 25 days × 9 individual samples = 225 individual samples per kind) in our study is enough for the E-nose experiments. Therefore, we think that the experiment in our study is workable and conformed to standards.
  2. As you said, there are always differences in the instrumental response due to the drift, which could be expected as e-noses are prone to drift. As we all known, the drift is a rather complex and inevitable effect in real scenarios, which is generated by different sources (such as sensor aging, sensor poisoning, thermal and memory effects of sensors, changes in environment and odor delivery system noise) [4]. So, strictly speaking, it is almost impossible to acquire a data set without any drift in real scenarios. In addition, the aim of our study is proposing a rapid and accurate method for the quality detection of base liquors and commercial liquors by means of a E-nose system in real scenarios. So, we conducted the experiments by independent measurements on individual samples in 25 days to satisfy the conditions of E-nose applications in real scenarios as much as possible. As a result, the distribution of experiment data in our study was closer to fact. PCA was used to distinguish the nine liquor samples as a regular tool in our study, and the results showed that PCA was not capable in this case. Therefore, we proposed a novel framework, namely ResNet-GBM, to mine the deep aroma features of the liquors for the quality detection of the nine liquor samples. The experimental results indicated that our proposed model presented good performances and strong robustness for the classification of the nine Chinese liquors, which shows a potential to avoiding the negative effect of the time drift in E-nose.
  3. In our study, the experiments were performed for 25 days to obtain more data, and the individual samples were measured to obtain more representative data. The data in the first 20 days was used for training (total 20 days × 9 individual samples = 180 individual samples per kind) and that in the last 5 days was used for test (total 5 days × 9 individual samples = 45 individual samples per kind). So, the training dataset and test dataset are independent, which can be used as a proper validation for the assessment of classification models.

References:

[1] Zhang, S., Cheng, Y., Luo, D., He, J., Wong, A. K., & Hung, K. (2021). Channel attention convolutional neural network for Chinese baijiu detection with E-nose. IEEE Sensors Journal, 21(14), 16170-16182.

[2] Li, X., Zhu, J., Li, C., Ye, H., Wang, Z., Wu, X., & Xu, B. (2018). Evolution of volatile compounds and spoilage bacteria in smoked bacon during refrigeration using an E-Nose and GC-MS combined with partial least squares regression. Molecules, 23(12), 3286.

[3] Xu, M., Wang, J., & Gu, S. (2019). Rapid identification of tea quality by E-nose and computer vision combining with a synergetic data fusion strategy. Journal of Food Engineering, 241, 10-17.

[4] Ziyatdinov, A., Marco, S., Chaudry, A., Persaud, K., Caminal, P., & Perera, A. (2010). Drift compensation of gas sensor array data by common principal component analysis. Sensors and Actuators B: Chemical, 146(2), 460-465.
